# Genomic Analysis of MRSA for Evaluating Local Transmission Dynamics in Geriatric Long-Term Care Facilities in Japan

**DOI:** 10.3390/antibiotics14090874

**Published:** 2025-08-30

**Authors:** Takayuki Suzuki, Teppei Sasahara, Shinya Watanabe, Koki Kosami, Dai Akine, Yumi Kinoshita, Longzhu Cui, Shuji Hatakeyama

**Affiliations:** 1Division of Infectious Diseases, Department of Infection and Immunity, Jichi Medical University, Yakushiji 3311-1, Shimotsuke 329-0498, Tochigi, Japan; takasuzuki-akt@jichi.ac.jp (T.S.); m00001da@jichi.ac.jp (D.A.); hatakeyamas@jichi.ac.jp (S.H.); 2Division of Bacteriology, Department of Infection and Immunity, Jichi Medical University, Yakushiji 3311-1, Shimotsuke 329-0498, Tochigi, Japan; swatanabe@jichi.ac.jp (S.W.); longzhu@jichi.ac.jp (L.C.); 3Division of Public Health, Center for Community Medicine, Jichi Medical University, Yakushiji 3311-1, Shimotsuke 329-0498, Tochigi, Japan; k.kosami@jichi.ac.jp; 4Department of Infectious Diseases, Ibaraki Prefectural Central Hospital, Koibuchi 6528, Kasama 309-1793, Ibaraki, Japan; 5Clinical Microbiology Laboratory, Jichi Medical University Hospital, Yakushiji 3311-1, Shimotsuke 329-0498, Tochigi, Japan; kinoshitayumi@jichi.ac.jp; 6Division of General Medicine, Center for Community Medicine, Jichi Medical University, Yakushiji 3311-1, Shimotsuke 329-0498, Tochigi, Japan

**Keywords:** methicillin-resistant *Staphylococcus aureus* (MRSA), whole genome sequencing, older adults, geriatric long-term care facility, infection control and prevention

## Abstract

**Background/Objectives:** Methicillin-resistant *Staphylococcus aureus* (MRSA) colonization in geriatric long-term care facilities (LTCFs) is a global concern. However, the transmission dynamics of MRSA among LTCF residents in Japan remain largely unknown. **Methods:** Whole-genome sequencing was conducted on 85 MRSA isolates obtained from 76 residents across 4 geriatric LTCFs in Japan. Single-nucleotide polymorphism (SNP) analysis was performed to identify the transmission dynamics, with a threshold of ≤15 pairwise core-genome SNP distances defining recent transmission clusters (genomic clusters). Antimicrobial susceptibility testing and investigation of antimicrobial resistance genes were also performed. **Results:** Among the 76 MRSA-carrying residents, 34 (44.7%) belonged to 14 genomic clusters, including strains from clinical specimens of 7 individuals. Three individuals acquired MRSA strains within the LTCFs, which were part of genomic clusters. Conversely, 14 residents who underwent testing immediately after admission carried MRSA strains within genomic clusters, suggesting transmission prior to their LTCF admission. MRSA isolates that were prevalent among LTCF residents were generally susceptible to trimethoprim–sulfamethoxazole but resistant to levofloxacin and clindamycin. **Conclusions:** Acquisition of MRSA genomic cluster strains among LTCF residents can occur both during and before admission to the facility. These findings underscore the need for measures that mitigate MRSA transmission inside and outside LTCFs.

## 1. Introduction

Antimicrobial resistance (AMR) poses a considerable global concern, with methicillin-resistant *Staphylococcus aureus* (MRSA) emerging as the second leading cause for AMR-associated mortality worldwide [1]. Residents of geriatric long-term care facilities (LTCFs) are particularly vulnerable to MRSA colonization or infection owing to prevalent risk factors, including advanced age, antimicrobial drug use, presence of wounds, nasogastric tube feeding, indwelling urinary catheter, and recent history of admission to acute care hospitals [2,3]. Several studies have documented greater incidences of MRSA colonization among residents of LTCFs compared with those in the general community and hospital settings [4,5]. Infection prevention and control (IPC) strategies have therefore become crucial for preventing MRSA transmission in LTCFs. Despite this, MRSA prevalence in geriatric LTCFs remained unchanged, even with IPC education and staff training [6]. Given that these institutions differ from acute care hospitals in terms of length of stay, required medical practice, assistance for daily living, and staff shortages, a direct application of hospital-based IPC strategies may be difficult to translate to the LTCF setting.

For the implementation of effective IPC strategies, understanding the MRSA transmission dynamics in LTCFs is essential. However, such data remain largely unknown in the context of Japan. While some studies have utilized whole-genome sequencing (WGS) to investigate MRSA transmission in LTCFs [7,8], there may be inherent differences in the circumstances of LTCF residents in Japan, such as the long-term care insurance system, provided services [9], and world-renowned aging populations, compared with those in previous studies [10]. Furthermore, the epidemiology of prevalent MRSA is subject to regional variation [3]. Previously, we investigated MRSA carriage among the residents of four LTCFs in Japan, revealing that MRSA carriage tended to be higher among those with shorter lengths of stay, although the difference was not statistically significant [11]. In this study, we aimed to determine the genomic epidemiology of MRSA transmission among residents in geriatric LTCFs in Japan by performing WGS to analyze the MRSA strains collected in the previous study.

## 2. Results

### 2.1. Genetic Features of MRSA Isolates

We discovered extensive genetic diversity among the isolates, representing multiple multi-locus sequence types (MLSTs), SCCmec, and spa types with varying proportions (Table 1). For LTCFs A, B, and D, dominant isolates with specific spa types were observed, revealing the predominance of community-associated MRSA (CA-MRSA) clones, such as ST1-SCCmec IV (ST1-IV), ST8-IV, ST22-IV, ST45-V, and ST72-IV. Conversely, healthcare-associated MRSA (HA-MRSA) clones, including ST5-II and ST764-II, and their single-locus variants were predominant in LTFC C. Notably, despite the adjacent location and comparable patient characteristics between LTCFs A and C, the most prevalent clones in each location varied.

### 2.2. SNP Analysis and Identification of Genomic Clusters

The presence of prevalent spa types within each LTCF suggests potential genomic clusters identified through single-nucleotide polymorphism (SNP) analysis. Excluding the single-strain CC22, we found that 34 (44.7%) MRSA-carrying residents of geriatric LTCFs belonged to 14 genomic clusters and that every LTCF had at least one genomic cluster (Figure 1A–D and Figure 2). Isolates of these clusters were detected from the clinical specimens, in addition to the nasal swabs, of seven individuals. Among the residents associated with genomic clusters, three residents (D-2, D-4, and D-11) acquired these strains within LTCF D. In contrast, 14 residents harbored strains prior to admission owing to positive initial testing results within 0.5 months of admission. While a cutoff value of 15 core-genome SNPs (cgSNPs) did not reveal any transmissions between LTCFs, JMUB7474 and JMUB7486, two strains that originated from residents of LTCFs A and C, respectively, were closely related, with a 33-cgSNP difference (Figure 2A). The former strain was isolated shortly after the individual’s admission, whereas the latter was detected 1.5 months later.

Moreover, to evaluate the temporal dynamics of MRSA carriage among LTCF residents, we reviewed alterations in the carrier status of residents who underwent initial testing on admission and a second test >2 months later (Figure 3). Among the 65 residents who met these criteria, 55 (84.6%) showed unchanged carrier status. Of the 14 MRSA carriers at admission, 7 (4.6%) became non-carriers at the second test, whereas 3 (2.9%) acquired genomic cluster strains following admission. However, before the results of SNP analysis were available, confirming whether the acquired MRSA strains originated within their LTCFs was not possible, as two of them were already carriers of other MRSA strains at the time of admission.

Pan-GWAS revealed that none of the 6667 potential coding sequences exhibited a significantly higher prevalence in genomic cluster strains.

### 2.3. Antimicrobial Susceptibility and Genetic AMR Profile

All isolates were susceptible to vancomycin, teicoplanin, daptomycin, linezolid, and mupirocin, and 84 (98.8%) were susceptible to trimethoprim–sulfamethoxazole. Conversely, 83 (97.6%) and 76 (89.4%) isolates were resistant to levofloxacin and clindamycin, respectively. Although 62 (72.9%) isolates were susceptible to minocycline, only 2 (28.6%) isolates in LTCF C were susceptible to minocycline. All minocycline-resistant isolates in LTCF C were ST764-II, and all ST764-II isolates in other LTCFs were also resistant to minocycline (Appendix A).

The molecular basis for levofloxacin resistance was determined to be the presence of at least one gyrA point mutation (S84L, S85P, E88G, or E88K) in 98.8% (82/83) of the isolates. Furthermore, all clindamycin-resistant isolates had either erm (A) or erm (C). All tetracycline-nonsusceptible isolates carried tet (M), which encodes a ribosomal protection protein (Appendix A).

### 2.4. Virulence Gene Profile

Overall, the number of toxic shock syndrome toxin-1 (tst-1) positive isolates was two (2.4%, Appendix A). In CC1, all ST1-IVa and ST2725-IVa isolates had sea, she, sek, seq, sak, and scn, excluding JMUB7519 (ST1-IVa). In CC5, all ST764-II isolates (*n* = 11) carried seb, seg, sei, sem, sen, seo, seu, sak, and scn. In CC8, there were only two USA300 clones (ST8-IVa with sek, seq, and lukF/S-PV).

## 3. Discussion

In this study, we elucidated the transmission dynamics of MRSA among residents in geriatric LTCFs in Japan, an aspect that was unclear in our previous study. Multiple putative transmission events were observed within each LTCF, with 34 (44.7%) MRSA-carrying residents sharing 14 individual genomic cluster strains. Although most residents showed no changes in carrier status, three residents acquired MRSA strains during LTCF admission. In addition, genomic cluster strains were discovered from the clinical specimens of seven individuals.

Our analysis showed that most genomic cluster strains among LTCF residents were likely acquired prior to admission. Owing to the absence of personal information, such as prior hospitalization, contact with healthcare environments, or household contact screening, it was not possible to determine the exact location of transmission. However, considering that most LTCFs cater to individuals requiring post-hospitalization rehabilitation therapy or daily living support beyond their home settings, hospitals emerge as the most likely source of external transmissions. This is consistent with existing evidence citing acute care hospitals as the most common source of inter-facility transmission [7]. However, outpatient care, including rehabilitation, can be provided as home-based services in Japan, allowing another avenue for transmissions to occur. Future research is required, as there is insufficient data on the spread of MRSA in patients undergoing outpatient care. Furthermore, to prevent MRSA transmission among LTCF residents, interventions inside and outside these facilities are paramount. Recent randomized controlled trials have shown that decolonizing MRSA carriers after hospital discharge and implementing universal decolonization among LTCF residents result in reduced MRSA infections and infection-related hospital transfers, respectively [12,13]. Therefore, the combined implementation of IPC strategies and decolonization approaches is crucial, particularly in areas where MRSA transmission is a clear concern. Our results provide molecular evidence for the necessity of interventions to reduce MRSA colonization among LTCF residents.

Further SNP analysis revealed that MRSA transmission among residents can cause both colonization and clinical infections. Of note, despite the prevalence of CA-MRSA clones, isolates prevalent within LTCFs displayed resistance to non-β-lactam agents, such as levofloxacin and clindamycin. Although CA-MRSA strains have exhibited resistance to multiple non-β-lactam agents in some cases [14], this specific drug susceptibility profile in LTCF residents remained unclear. Our findings suggest that most of the prevalent MRSA strains were susceptible to trimethoprim–sulfamethoxazole. However, considering the high prevalence of renin–angiotensin system inhibitor (RASi) usage for hypertension among LTCF residents [15], cautious administration is warranted, as concurrent trimethoprim–sulfamethoxazole and RASi use has been associated with an increased risk of sudden mortality [16]. This underscores the balance of minimizing MRSA transmission among LTCF residents with consideration of other comorbidities.

Similarities and differences were observed in the characteristics of MRSA strains across LTCFs. Notably, despite the geographical proximity and shared hospital transfers, LTCFs A and C showed significant variation in the prevalence of specific ST-SCC*mec* types. This is particularly interesting as LTCF C is designed to assist with activities of daily living, whereas LTCF A prioritizes rehabilitation for the return of individuals to their homes. Such differences may reflect unique resident profiles, including the levels of required medical practice, length of stay, and degrees of exposure to other residents and healthcare personnel. These factors may have influenced the acquisition of distinct clones. Another factor to consider is that strains prevalent within the region may affect each facility. In our study, ST8-IV isolates were predominant in LTCF D but not in other facilities. A nationwide analysis of MRSA isolated from bloodstream infections in Japan revealed that ST8-IV was the most prevalent strain in areas where LTCF D was located (49.0%) but accounted for only 23.2% in areas where LTCFs A, B, and C were located [17]. This suggests that both facility characteristics and local endemic strains influence MRSA prevalence among LTCF residents. Additionally, ST764-II isolates were found in three LTCFs situated in both eastern and western parts of Japan. Recently, ST764-II has been identified as a high-risk clone, with a 30-day mortality rate of 48% for bloodstream infections [18]. This clone has diverged from the New York/Japan clone (ST5-II) and is characterized by a high proportion of seb. Although ST764-II clones were reported to have spread in the Japanese LTCF environment, primarily in the Chubu region in 2019 [19], the LTCFs in our study were located in different areas, suggesting that this high-risk clone has widely disseminated across LTCFs in Japan. Given that most LTCF residents have comorbidities that could complicate their clinical course, it may be necessary to closely monitor whether this emerging clone will continue to grow.

Despite the insights offered by our study, several limitations should be acknowledged. First, valuable information, such as the residents’ age, sex, underlying medical conditions, and occupied rooms, was unavailable in our prior study. As such, we could not investigate the transmissions between residents based on potential contact patterns. Second, our sample size was relatively smaller than those of previous studies [7,8]. Third, sampling practices varied across facilities, including the frequency of screening tests, timing of initial testing, and duration of study. These differences make it difficult to encompass the temporal aspects of transmission. Fourth, only nasal samples were collected. Since MRSA clonal complexes have a predilection towards either the nares or groin [20], and the oral cavity is recognized as a significant reservoir source [21,22], samples from other body sites could reveal unidentified transmission cases. Fifth, we examined only one colony from each individual. Multiple colonies could also demonstrate unidentified transmission cases. Lastly, staff and visitor screening was not performed, and environmental samples were not collected. With these samples, a more comprehensive evaluation of the transmission pathways could be possible. However, because these limitations hindered the identification of transmission instances, overestimation is unlikely in this study. Future studies should focus on gathering detailed resident information, including prior hospitalizations and transfers between LTCFs, longitudinal sampling, and contact tracing, including household members and LTCF staff, to more definitively distinguish between MRSA acquisition sites inside and outside LTCFs. A thorough understanding of transmission dynamics could elucidate unknown targets for focused instruction of standard precautions and decolonization of populations such as outpatient rehabilitation care.

## 4. Materials and Methods

### 4.1. Bacterial Isolates and Study Facilities

WGS was performed using MRSA isolates obtained from four geriatric LTCFs—anonymized as A, B, C, and D (Table 2)—in Japan; the isolate collection has been published previously [11]. Briefly, these MRSA isolates were collected for a cohort study using Seed-Swab MRSA™ kits (Eiken Chemical Co., Ltd., Tokyo, Japan) and cultured on MRSA selective agar plates (MDRS-K™; Kyokuto Pharmaceutical Industrial Co., Ltd., Tokyo, Japan). Microbial identification was confirmed using a VITEK 2 automated system (bioMerieux, Durham, NC, USA), and oxacillin resistance was determined using the disk diffusion method. A single colony from each individual resident was investigated. LTCFs A and C were adjacent, and residents of these facilities may be transferred to the same hospital. MRSA strains were isolated between 1 August 2018 and 31 March 2020. In addition to nasal samples, MRSA isolates were collected from specimens obtained from infectious sites for diagnostic purposes during the specified period (clinical isolates). Among the 453 residents in the LTCFs, 85 MRSA isolates were obtained from 76 residents, including 9 clinical isolates obtained from 8 residents. Notably, LTCF D had the highest number of MRSA-carrying residents (*n* = 44).

### 4.2. WGS and Transmission Analysis

WGS was performed for all collected isolates to investigate MRSA transmission among LTCF residents, where SNP analysis determined the transmission events. Genomic DNA was extracted using the QIAamp DNA mini kit (Qiagen, Hilden, Germany), and paired-end libraries were prepared using Nextera XT Library Prep (Illumina, Inc., San Diego, CA, USA). Sequencing was performed with the Illumina HiSeq XTen system at Macrogen (Illumina, Inc., San Diego, CA, USA) or the Illumina MiSeq (Illumina, Inc., San Diego, CA, USA) at our university using the MiSeq Reagent Kit v.3 (Illumina, Inc., San Diego, CA, USA). The resulting data have been deposited under accession number DRA017333. The individual accession numbers of the isolates are listed in Appendix A.

We performed trimming of raw reads to eliminate adapters and low-quality sequences, followed by de novo assembly using CLC Genomics Workbench version 9 (CLCbio, Qiagen, Valencia, CA, USA). We determined the MLST, SCC*mec* type, and *spa* type for each isolate using PubMLST, SCCmecFinder v1.2, and spaTyper v1.0 [23,24,25,26,27]. The core genome of each clonal complex was identified using Prokka v1.14.6 [28] and Roary v3.13.0 [29]. The pairwise cgSNP distance matrix was calculated using pairsnp v0.3.1 [30], and isolates sharing a cgSNP distance ≤15 were defined as genomic clusters, which is indicative of recent transmission [31]. The genomes AP018923.1, CP091525.1, AP020318.1, and CP083259.1 were used as references for CC1, CC5, CC8, and CC45, respectively. To determine the reference genomes, genome data of *S. aureus* with an assembly level of complete genome or chromosome as of 5 October 2023 were collected using NCBI genome download. The genomic data with the closest genomic distance to our strain were determined as the reference genome for each CC using Mash v2.3 [32]. Maximum-likelihood phylogenetic trees were then generated using a general time reversible model and gamma correction, followed by visualization with metadata [33,34,35].

To assess the temporal dynamics of MRSA carriage among LTCF residents, changes in the carrier status were compared between the initial test on admission and a second test >2 months later. The transition of MRSA carriage status was visually represented using a Sankey diagram created with the plotly library version 5.18.0 and Python 3.10.8. Additionally, a pan-genome-wide association study (pan-GWAS) was performed using Scoary v1.6.16 to investigate bacterial factors influencing transmission [36].

### 4.3. Antimicrobial Susceptibility Testing and Detecting AMR Genes

Antimicrobial susceptibility testing was performed using the BD Phoenix M50 system with the PMIC/ID-86 panel (BD Diagnostic Systems, Franklin Lakes, NJ, USA). The results were interpreted in accordance with the Clinical and Laboratory Standards Institute (CLSI) M100 33rd edition guidelines [37]. Furthermore, AMR genes, point mutations, and disinfectant-resistance genes were identified using ResFinder v4.3.3 [24,38,39]. The threshold for gene detection was set to 90% identity and 80% coverage.

### 4.4. Detection of Virulence Factor

Virulence genes were identified using VirulenceFinder v2.0.4 [40,41]. The threshold for gene detection was set to 90% identity and 80% coverage.

## 5. Conclusions

Our study showed that MRSA acquisition among residents of geriatric LTCFs can occur within the facility and prior to admission. This underscores the need for effective interventions within and outside LTCFs to prevent MRSA transmission in this cohort of individuals.

## Figures and Tables

**Figure 1 antibiotics-14-00874-f001:**
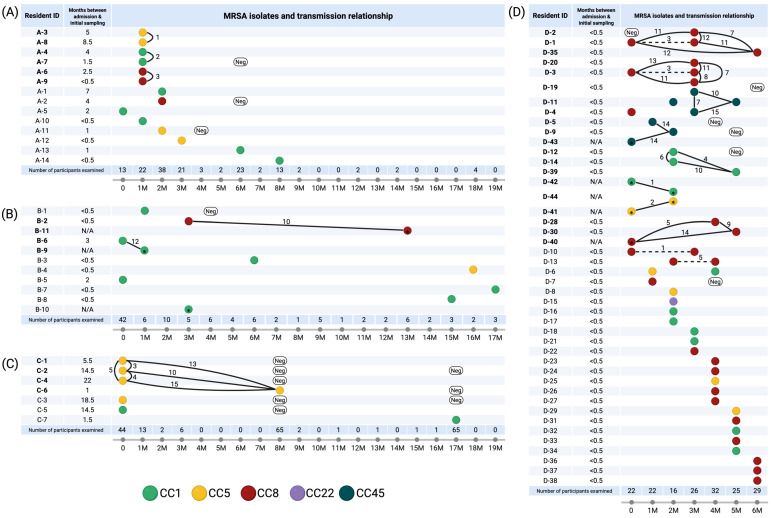
Methicillin-resistant Staphylococcus aureus (MRSA) transmission dynamics in four long-term care facilities. (**A**–**D**) show MRSA genomic clusters in long-term care facilities A, B, C, and D, respectively. Each circle represents an MRSA isolate, and the strains of clonal complex (CC) 1, CC5, CC8, CC22, and CC45 are indicated using green, yellow, red, purple, and indigo colors, respectively. Isolates from clinical specimens are marked with a * in circles. Samples with a core genome single-nucleotide polymorphism (cgSNP) distance ≤15 were connected by a solid line if they originated from different hosts and by a dashed line if they originated from the same host. The numbers attached to these lines indicate the cgSNP distance. The bottom axis is the time axis, where 0 denotes the month in which this study began at each institution, and the unit of the number is measured in months (M). This figure was created in BioRender. Suzuki, T. (2025) https://BioRender.com/biq47q2 (accessed on 5 July 2025). Abbreviations: CC, clonal complex; Neg, negative.

**Figure 2 antibiotics-14-00874-f002:**
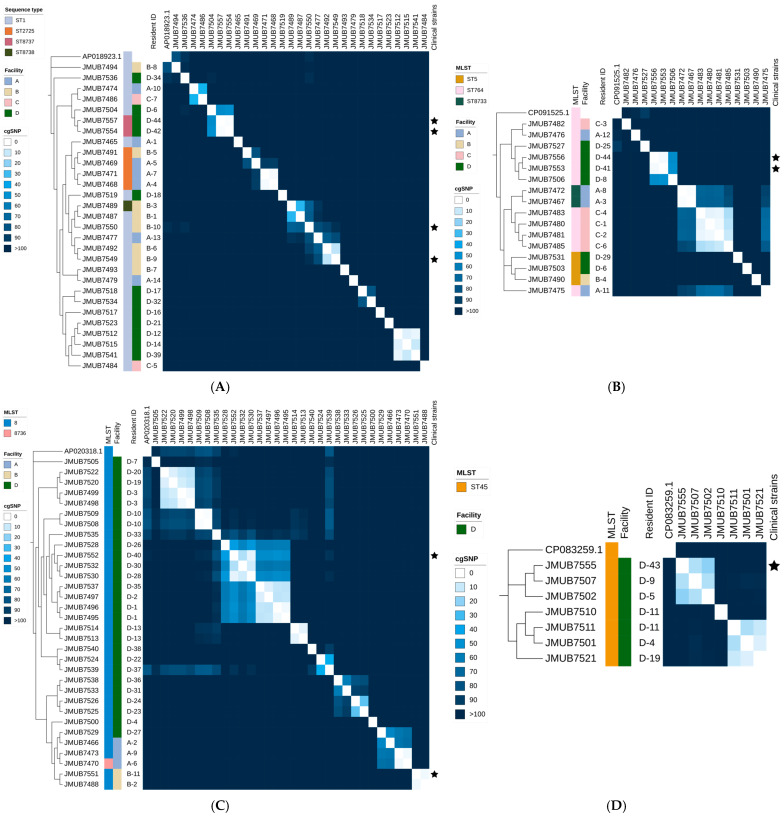
Maximum-likelihood phylogenetic trees with heatmaps based on core-genome single-nucleotide polymorphism (cgSNP) analysis. (**A**–**D**) depict clonal complexes (CC) 1, CC5, CC8, and CC45, respectively. The reference genomes of each clonal complex are shown as outgroups. To the right of the sample name, the multi-locus sequence type (MLST), facility, patient ID for this study, heatmap displaying the cgSNP distance, and clinical specimens are presented. Isolates from clinical specimens are marked with ★ (star). Abbreviations: MLST, multi-locus sequence type; cgSNP, core-genome single-nucleotide polymorphism.

**Figure 3 antibiotics-14-00874-f003:**
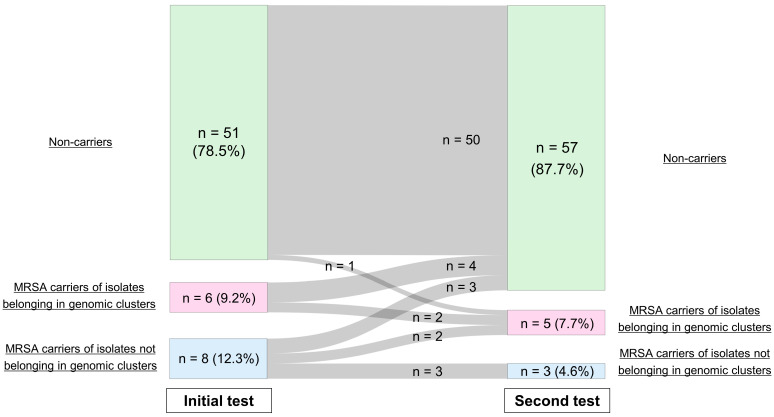
Temporal changes in the methicillin-resistant *Staphylococcus aureus* (MRSA)-carrying state. Residents who underwent an initial test immediately after admission and a follow-up test >2 months later were included in this analysis. For each test, they were classified as MRSA non-carriers, MRSA carriers of isolates involved in genomic clusters, and MRSA carriers of isolates not involved in genomic clusters. Only three participants (4.6%) acquired genomic cluster strains after admission, whereas seven MRSA carriers became non-carriers in the second test.

**Table 1 antibiotics-14-00874-t001:** Distribution of isolate types at each facility.

			A	B	C	D
**CC and MLST, *n* (%)**						
	CC1					
		ST1	4 (28.6%)	6 (54.5%)	2 (28.6%)	10 (18.9%)
		ST2725	3 (21.4%)	1 (9.1%)	-	-
		ST8737	-	-	-	2 (3.8%)
		ST8738	-	1 (9.1%)	-	-
	CC5					
		ST5	-	1 (9.1%)	-	2 (3.8%)
		ST764	2 (14.3%)	-	5 (71.4%)	4 (7.5%)
		ST8733	2 (14.3%)	-	-	-
	CC8					
		ST8	2 (14.3%)	2 (18.2%)	-	27 (50.9%)
		ST8736	1 (7.1%)	-	-	-
	CC22					
		ST22	-	-	-	1 (1.9%)
	CC45					
		ST45	-	-	-	7 (13.2%)
SCC*mec* type, *n* (%)						
	II		4 (28.6%)	1 (9.1%)	5 (71.4%)	7 (13.2%)
	IVa		10 (71.4%)	8 (72.7%)	2 (28.6%)	13 (24.5%)
	IVc		-	2 (18.2%)	-	-
	IVj		-	-	-	27 (50.9%)
	V		-	-	-	6 (11.3%)
*spa* type, *n* (%)						
	t002		3 (21.4%)	-	3 (42.9%)	1 (1.9%)
	t005		-	-	-	1 (1.9%)
	t008		2 (14.3%)	2 (18.2%)	-	2 (3.8%)
	t010		-	-	1 (14.3%)	-
	t045		1 (7.1%)	-	-	3 (5.7%)
	t062		-	-	-	1 (1.9%)
	t211		-	-	-	3 (5.7%)
	t334		-	-	-	1 (1.9%)
	t1062		-	1 (9.1%)	-	-
	t1081		-	-	-	6 (11.3%)
	t1767		-	-	-	2 (3.8%)
	t1781		-	-	1 (14.3%)	-
	t1784		7 (50.0%)	5 (45.5%)	2 (28.6%)	11 (20.8%)
	t2427		-	-	-	1 (1.9%)
	t3081		-	-	-	2 (3.8%)
	t4494		-	3 (27.3%)	-	-
	t5071		-	-	-	15 (28.3%)
	t7744		-	-	-	1 (1.9%)
	t20946		1 (7.1%)	-	-	-
	Unknown type		-	-	-	3 (5.7%)

Abbreviations: CC, clonal complex; MLST, multi-locus sequence type; ST, sequence type; SCC*mec*, staphylococcal cassette chromosome *mec*.

**Table 2 antibiotics-14-00874-t002:** Characteristics of participants and isolates at each facility.

			A	B	C	D
Facility type			HSF	HSF	SNH	HSF
Resident capacity, *n*			100	50	60	150
Region of Japan			Eastern	Eastern	Eastern	Western
Number of participants, n			128	99	89	135
Number of participants with at least one MRSA isolation, *n*			14	11	7	44
Number of isolates, *n*			14	11	7	53
Source of samples, *n*						
	Nasal swab					
	Clinical specimen					
		Airway tract secretions	-	-	-	5
		Urine	-	-	-	1
		Otitis externa	-	1	-	-
		Skin abscess	-	1	-	-
		Dacryocystitis	-	1	-	-
Number of participants sorted by test frequency, n						
	1 time		109	90	18	96
	2 times		19	9	30	39
	3 times		-	-	41	-
Frequency of nasal swab tests, median (IQR)			1 (1–1)	1 (1–1)	2 (2–3)	1 (1–2)
Timing of the first nasal swab test from admission, median (IQR), month			2.5 (1.0–7.4)	<0.5 (<0.5–4.3)	7.0 (1.0–25.5)	<0.5 (<0.5–<0.5)
Duration of the study, month			20	18	20	7

Abbreviations: HSF, health services facility; SNH, special nursing home; MRSA, methicillin-resistant *Staphylococcus aureus*; IQR, interquartile range.

## Data Availability

The whole-genome data from this study were deposited in DDBJ/ENA/GenBank under the BioProject accession number PRJDB16917.

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
