# Peer review of "Genomic Analysis of MRSA for Evaluating Local Transmission Dynamics in Geriatric Long-Term Care Facilities in Japan"

_antibiotics, 2025, doi:10.3390/antibiotics14090874_

Round 1

Reviewer 1 Report

Comments and Suggestions for Authors

Reviewer Comments to the Authors:

The manuscript titled "Genomic analysis of MRSA for evaluating local transmission dynamics in geriatric long-term care facilities in Japan" is a well-conducted and highly relevant study. The use of whole-genome sequencing to investigate MRSA transmission dynamics among LTCF residents offers valuable insights into both facility-level and community-level dissemination patterns. The Discussion section, in particular, is thoughtful and reflective of the study’s limitations and strengths.

I would like to offer the following minor suggestions to further strengthen the manuscript:

  1. Interpretation in the Results Section (lines 93–100):

Some of the statements in the Results section, such as "indirect transmission may have occurred" and the discussion of transmission routes between LTCFs, involve interpretation beyond the descriptive data. To maintain clarity and adhere to standard scientific reporting, it may be helpful to relocate such comments to the Discussion section, where their speculative nature can be more appropriately addressed.

  1. Study Limitations and Opportunities for Future Research:

The manuscript commendably acknowledges several important limitations, such as the lack of serial MRSA testing for all residents, absence of environmental or staff screening, and limited information on residents’ healthcare histories prior to LTCF admission. These candid reflections strengthen the credibility of the study. Building on this transparency, the authors may consider briefly outlining how future studies could address these gaps—such as integrating longitudinal sampling, contact tracing (including household members), or data on previous hospitalizations—to more definitively distinguish between community- and facility-acquired MRSA. Even if such approaches were not feasible in the present study, highlighting them could inspire further work in this important area. This suggestion stems from genuine scientific interest and appreciation for the scope and depth of the current research.

3. Clarification on the Source of MRSA at Admission:

Several strains belonging to genomic clusters were detected shortly after resident admission. While the manuscript acknowledges the challenge in distinguishing between community-acquired and facility-acquired MRSA due to the lack of serial testing, it remains a critical question whether further investigations—such as prior hospitalization history, contact with healthcare environments, or household contact screening—were considered or could be explored in future studies. Even if such data were not available for the current analysis, a brief reflection on this possibility could enrich the discussion and stimulate future directions. I raise this not as a criticism, but rather out of scientific curiosity and appreciation for the depth of the study.

Overall, this is a well-written manuscript with minimal revisions required. I commend the authors for their careful analysis and thoughtful interpretation.

Author Response

Comments 1: Interpretation in the Results Section (lines 93–100): Some of the statements in the Results section, such as "indirect transmission may have occurred" and the discussion of transmission routes between LTCFs, involve interpretation beyond the descriptive data. To maintain clarity and adhere to standard scientific reporting, it may be helpful to relocate such comments to the Discussion section, where their speculative nature can be more appropriately addressed.

Response 1: Thank you for your valuable comments. We have made revisions to our manuscript in order to present only descriptive findings in the Results section as follows:

“While a cutoff value of 15 core-genome SNPs (cgSNPs) did not reveal any transmissions between LTCFs, JMUB7474 and JMUB7486, two strains that originated from residents of LTCFs A and C, respectively, were closely related, with a 33-cgSNP difference (Figure 2A). The former strain was isolated shortly after the individual's admission, whereas the latter was detected 1.5 months later.” (line 93-98)

Comments 2: Study Limitations and Opportunities for Future Research: The manuscript commendably acknowledges several important limitations, such as the lack of serial MRSA testing for all residents, absence of environmental or staff screening, and limited information on residents’ healthcare histories prior to LTCF admission. These candid reflections strengthen the credibility of the study. Building on this transparency, the authors may consider briefly outlining how future studies could address these gaps—such as integrating longitudinal sampling, contact tracing (including household members), or data on previous hospitalizations—to more definitively distinguish between community- and facility-acquired MRSA. Even if such approaches were not feasible in the present study, highlighting them could inspire further work in this important area. This suggestion stems from genuine scientific interest and appreciation for the scope and depth of the current research.

Response 2: We appreciate your important comments. We have included how future studies could overcome our research constraints as follows:

“Future studies should focus on gathering detailed resident information including prior hospitalizations and transfers between LTCFs, longitudinal sampling, contact tracing including household members and LTCFs staff, to more definitively distinguish between MRSA acquisition sites inside and outside LTCFs. A thorough understanding of transmission dynamics could elucidate unknown target for focused instruction of standard precaution and decolonization population such as outpatient rehabilitation care.” (line 233-238)

Comments 3: Clarification on the Source of MRSA at Admission: Several strains belonging to genomic clusters were detected shortly after resident admission. While the manuscript acknowledges the challenge in distinguishing between community-acquired and facility-acquired MRSA due to the lack of serial testing, it remains a critical question whether further investigations—such as prior hospitalization history, contact with healthcare environments, or household contact screening—were considered or could be explored in future studies. Even if such data were not available for the current analysis, a brief reflection on this possibility could enrich the discussion and stimulate future directions. I raise this not as a criticism, but rather out of scientific curiosity and appreciation for the depth of the study.

Response 3: We appreciate the reviewer’s constructive feedback. We have revised to our manuscript in second paragraph of the Discussion section as follows:

“Our analysis showed that most genomic cluster strains among LTCF residents were likely acquired prior to admission. Owing to the absence of personal information, such as prior hospitalization, contact with healthcare environments, or household contact screening, it was not possible to determine the exact location of transmission. However, considering that most LTCFs cater to individuals requiring post-hospitalisation rehabilitation therapy or daily living support beyond their home settings, hospitals emerge as the most likely source of external transmissions. This is consistent with existing evidence citing acute care hospitals as the most common source of inter-facility transmission [7].” (line 162-169)

Reviewer 2 Report

Comments and Suggestions for Authors

Thank you for the opportunity to review this manuscript that describes results of WGS of MRSA isolates obtained from LTCF residents in Japan.

The analysis of the isolates is extensive, and the authors have spent a lot of effort determining sequence types and classifications of the isolates as well as relatedness. 

However, there is a high degree of variation between LTCFs, making the amount of data to consume difficult. Additionally, by their own admission in the limitations, there is an insufficient number of isolates collected in an inconsistent way without accompanying epidemiological data. Without proper timing and exposure data, it is impossible to say where the transmission occurred, and the authors and readers are left to speculate. Taken together, it is hard to conclude anything from this study other than MRSA person-to-person transmission is possible, which is not a novel finding. 

Author Response

Reviewer #2’s comments

Thank you for the opportunity to review this manuscript that describes results of WGS of MRSA isolates obtained from LTCF residents in Japan. 

The analysis of the isolates is extensive, and the authors have spent a lot of effort determining sequence types and classifications of the isolates as well as relatedness.

However, there is a high degree of variation between LTCFs, making the amount of data to consume difficult. Additionally, by their own admission in the limitations, there is an insufficient number of isolates collected in an inconsistent way without accompanying epidemiological data. Without proper timing and exposure data, it is impossible to say where the transmission occurred, and the authors and readers are left to speculate. Taken together, it is hard to conclude anything from this study other than MRSA person-to-person transmission is possible, which is not a novel finding.

Response: We thank the reviewer for the careful review of the manuscript. We appreciate the reviewer's comment regarding the high degree of variation among LTCFs and the lack of clinical epidemiology data. We understand that these factors make it difficult to draw definitive conclusions about the timing and location of transmission, but we believe it is an important step. Our main goal was to use whole-genome sequencing to understand the molecular epidemiology of MRSA in Japanese LTCFs. The identification of genomic clusters serves as a strong indicator of potential transmission events and provides a foundation for future, more comprehensive studies that could include detailed epidemiological data.

Reviewer 3 Report

Comments and Suggestions for Authors

The manuscript by Suzuki et al. presents a genomic epidemiological study of MRSA transmission in Japanese geriatric long-term care facilities using whole-genome sequencing. The findings highlight that MRSA acquisition occurs both before and during LTCF stay, underscoring the need for targeted infection control measures.

Overall, the manuscript is well-motivated and methodologically sound, and it provides valuable insights for tailoring infection prevention strategies specific to LTCFs. However, several aspects could benefit from further clarification or expansion. Taking these points and the comments below into consideration, we recommend the manuscript for publication in your journal following minor revisions:

  1. Could you elaborate on the rationale behind using a ≤15 SNP threshold to define genomic clusters? Is this threshold specific to Staphylococcus aureus, or is it a broadly accepted standard across bacterial genomic epidemiology?
  2. Are there plans to incorporate staff, visitor, or environmental sampling in future studies to achieve a more comprehensive understanding of MRSA transmission dynamics within LTCFs?
  3. Your conclusion highlights the need for interventions both within and beyond LTCFs. What specific infection control strategies would you consider most effective at each level?
  4. What are the anticipated next steps for translating these genomic insights into actionable public health measures or clinical guidelines aimed at reducing MRSA transmission in long-term care settings?

Author Response

Comments 1: Could you elaborate on the rationale behind using a ≤15 SNP threshold to define genomic clusters? Is this threshold specific to Staphylococcus aureus, or is it a broadly accepted standard across bacterial genomic epidemiology?

Response 1: We appreciate your valuable comments. In general, the SNP threshold to define closely related isolates depends on factors such as the size of the pathogen's genome, its mutation rate, and the context (with mutation rates being higher during outbreaks compared to non-outbreak situations) [1]. Coll F et al. conducted a retrospective genomic and epidemiological study of MRSA in the United Kingdom, revealing that a cutoff of 15 cgSNPs can exclude MRSA transmission within the past six months [2]. Consequently, we employed the same methodology to determine cgSNP distances, using a threshold of ≤15 SNPs to define genomic clusters. This threshold is specific to MRSA; for instance, a ≤17 SNP threshold is used for defining genomic clusters in Escherichia coli [3]. As noted in lines 275-276 of the Method section, we explained the reasoning for this threshold: “isolates sharing a cgSNP distance ≤15 were defined as genomic clusters, which is indicative of recent transmission [31].”

<References>

  1. Hoffman, S.; Lapp, Z.; Wang, J.; Snitkin, E.S. Regentrans: A Framework and R Package for Using Genomics to Study Regional Pathogen Transmission. Microb. Genom. 2022, 8, doi:10.1099/mgen.0.000747.
  2. Coll, F.; Raven, K.E.; Knight, G.M.; Blane, B.; Harrison, E.M.; Leek, D.; Enoch, D.A.; Brown, N.M.; Parkhill, J.; Peacock, S.J. Definition of a Genetic Relatedness Cutoff to Exclude Recent Transmission of Meticillin-Resistant Staphylococcus Aureus: A Genomic Epidemiology Analysis. Lancet Microbe 2020, 1, e328–e335.
  3. Ludden, C.; Coll, F.; Gouliouris, T.; Restif, O.; Blane, B.; Blackwell, G.A.; Kumar, N.; Naydenova, P.; Crawley, C.; Brown, N.M.; et al. Defining Nosocomial Transmission of Escherichia Coli and Antimicrobial Resistance Genes: A Genomic Surveillance Study. Lancet Microbe 2021, 2, e472–e480.

Comments 2: Are there plans to incorporate staff, visitor, or environmental sampling in future studies to achieve a more comprehensive understanding of MRSA transmission dynamics within LTCFs?

Response 2: Yes. We acknowledge that the absence of sampling from staff, visitors, and environment is a limitation to completely understand the dynamics of MRSA acquisition. We have added the following future perspectives:

“Future studies should focus on gathering detailed resident information including prior hospitalizations and transfers between LTCFs, longitudinal sampling, contact tracing including household members and LTCFs staff, to more definitively distinguish between MRSA acquisition sites inside and outside LTCFs.” (line 233-236)

Comments 3: Your conclusion highlights the need for interventions both within and beyond LTCFs. What specific infection control strategies would you consider most effective at each level?

Response 3: We thank the reviewer for these insightful comments. Since the objective of this study was to elucidate the molecular epidemiology of MRSA isolated from elderly residents in Japanese LTCFs, it is difficult to determine which infection control strategies are most effective inside and outside LTCFs. Nevertheless, potential infection prevention strategies include the comprehensive implementation of standard precautions and contact precautions, as well as the performance of decolonization on the appropriate patient groups. In particular, decolonization of hospitalized patients carrying MRSA after discharge or universal decolonization targeting nursing home residents has been validated in randomized controlled trials [4,5] (see line 174-177). We expect that intervention targets outside of hospitals and LTCFs will be identified by future research that aims to better understand transmission dynamics.

<References>

  1. Huang, S.S.; Singh, R.; McKinnell, J.A.; Park, S.; Gombosev, A.; Eells, S.J.; Gillen, D.L.; Kim, D.; Rashid, S.; Macias-Gil, R.; et al. Decolonization to Reduce Postdischarge Infection Risk among MRSA Carriers. N. Engl. J. Med. 2019, 380, 638–650.
  2. Miller, L.G.; McKinnell, J.A.; Singh, R.D.; Gussin, G.M.; Kleinman, K.; Saavedra, R.; Mendez, J.; Catuna, T.D.; Felix, J.; Chang, J.; et al. Decolonization in Nursing Homes to Prevent Infection and Hospitalization. N. Engl. J. Med. 2023, 389, 1766–1777.

Comments 4: What are the anticipated next steps for translating these genomic insights into actionable public health measures or clinical guidelines aimed at reducing MRSA transmission in long-term care settings?

Response 4: Thank you for your important comments. We have revised to our manuscript to add future perspectives as follows:

“Future studies should focus on gathering detailed resident information including prior hospitalizations and transfers between LTCFs, longitudinal sampling, contact tracing including household members and LTCFs staff, to more definitively distinguish between MRSA acquisition sites inside and outside LTCFs. A thorough understanding of transmission dynamics could elucidate unknown target for focused instruction of standard precaution and decolonization population such as outpatient rehabilitation care.” (line 233-238)